# Using colony size to measure fitness in *Saccharomyces cerevisiae*

**James H. Miller[1], Vincent J. Fasanello[2], Ping Liu[2], Emery R. Longan[1], Carlos A. Botero[2¤], Justin C. Fay** [1] *

**1** Department of Biology, University of Rochester, Rochester, New York, United States of America,
**2** Department of Biology, Washington University, St. Louis, Missouri, United States of America

¤ Current address: Department of Integrative Biology, University of Texas at Austin, Austin, Texas, United States of America
* justin.fay@rochester.edu

**Data Availability Statement:** All the raw and processed data, analysis scripts and data tables are available at OSF: https://osf.io/ewz8x/. Data underlying each figure is available at the same location. A list of reagents, equipment and software is also provided in S3 Table.

## Abstract

Competitive fitness assays in liquid culture have been a mainstay for characterizing experimental evolution of microbial populations. Growth of microbial strains has also been extensively characterized by colony size and could serve as a useful alternative if translated to per generation measurements of relative fitness. To examine fitness based on colony size, we established a relationship between cell number and colony size for strains of *Saccharomyces cerevisiae* robotically pinned onto solid agar plates in a high-density format. This was used to measure growth rates and estimate relative fitness differences between evolved strains and their ancestors. After controlling for edge effects through both normalization and agar-trimming, we found that colony size is a sensitive measure of fitness, capable of detecting 1% differences. While fitnesses determined from liquid and solid mediums were not equivalent, our results demonstrate that colony size provides a sensitive means of measuring fitness that is particularly well suited to measurements across many environments.

## Introduction

Fitness is a measure of an organism's ability to survive and reproduce. Because evolution is driven by fitness differences in a population, measuring fitness is important to identifying phenotypes and genotypes that evolve due to natural selection and determining a population's response to selection and the rate of adaptation. However, measuring fitness can be difficult. Fitness is sensitive to environmental conditions [1, 2], the timescale over which it is measured [3, 4], and can be frequency or density dependent [5, 6]. Furthermore, differences in fitness as small as the reciprocal of the effective population size can be important in evolution [7, 8]. Laboratory-based fitness assays offer control over environmental conditions and population parameters, and have thus been extensively used to characterize fitness [9, 10].

Competitive fitness assays have been the most widely used laboratory method for quantifying fitness. Competitive fitness is measured by the change in the frequencies of two or more strains grown in the same environment where they compete for the same resources. These changes are measured by quantifying strain-specific fluorescence, antibiotic resistance, or

**Funding:** This work was supported by a National Institutes of Health grant (GM080669) to JCF. The funder had no role in study design, data collection and analysis, decision to publish, or preparation of the manuscript.

**Competing interests:** The authors have declared that no competing interests exist.

DNA barcode markers over time. While fluorescence-based quantitation is often limited to single competitions, the use of barcodes enables multiplexing within pooled fitness assays, e.g. [11, 12]. However, competitive fitness assays often assume that the relative fitness of two strains is the same as that obtained from each compared to a common reference, and the assumption of transitive fitness is not always valid [13–15]. Competitive fitness assays can also be hard to scale to multiple environments as each condition requires a separate assay. Measuring fitness across environments is important to assessing selection gradients, fitness trade-offs, and the evolution of specialist versus generalists [16–18].

Fitness can also be measured using individual growth assays. In these non-competitive assays a single strain is arrayed across multiple environments and growth can be measured by optical density in liquid cultures in microtiter plates or by changes in colony intensity or size on solid agar plates [19]. Much progress has been made in developing sensitive and robust measures of growth over time [20–31]. This has been accomplished through accurate measurement of cell number from optical density or colony size, quantifying growth parameters such as the lag time, growth rate and carrying capacity, and normalization procedures to account for spatial position effects. However, because growth differences are not translated into per generation measures of fitness and corresponding selection coefficients [32], it is hard to know whether colony size assays are sensitive enough to detect small differences (~1%) in relative fitness that are often measurable using competitive growth assays. For example, given two populations of equal size that have a 1% fitness difference (per generation), the ratio of the two population sizes is expected to be 0.990 after one generation but 0.905 after 10 generations under a continuous time model [32]. One notable exception is the estimation of competitive fitness based on the spatial sector patterns of colonies produced from a mixture of marked strains [33].

In this study we evaluate per generation fitness measurements derived from colony size on agar plates and compare them to competitive fitness assays. Colonies were robotically pinned in a high density (384) format and colony size was translated to cell number, which we then used to calculate Malthusian growth rates and relative fitness differences. We evaluated both normalization and agar trimming as means of eliminating spatial position effects and assessed the sensitivity and reproducibility of the fitness measurements. Using this setup, we measured the fitness of six evolved strains in comparison to their ancestors across a range of stress conditions. Compared to competitive fitness measurements, we find that fitness based on colony size can be just as sensitive but is more amenable to highly replicated measures of fitness across many environments.

## Methods

### Strains

Six evolved strains were obtained from single colony isolates of six evolved populations that were generated as part of a larger study [34] (S1 Table). In the study, each population was started using two barcoded diploid derivatives of an oak tree strain. Populations were evolved in 0.5 mL liquid cultures in 96 well plates that were passaged daily for 50 days, resulting in ~500 generations of experimental evolution. Cultures were grown at 30°C in complete medium (CM; 20 g/l dextrose, 1.7 g/l yeast nitrogen base without amino acid and ammonium sulfate, 5.0 g/l ammonium sulfate, 1.3 g/l dropout mix complete without yeast nitrogen base), supplemented with either 80% the lethal limit of $CuSO_4$ (6.4 μM), NaCl (274 mM), or neither. The six resulting evolved populations had an evolutionary history (EH) of no stress (EH0), constant stress at 80% of the lethal limit of either sodium or copper (EH80), or daily fluctuations between the two (EH0_80). Because each of the evolved strains had two potential

ancestors, both potential ancestors ("ancestor pair") were used to determine fitness (S1 Table). We generated a YFP-expressing strain [35] with the same genetic background as the ancestor strains (YJF4604) for competitive fitness assays.

## Colony size assay

Fitness differences between evolved strains and their respective ancestor pairs were determined from growth rate differences on solid agar CM plates with varying concentrations of copper (CuSO4) or sodium (NaCl) stress (S2 Table). Single colony replicates of each strain were grown overnight in 1 mL rich medium (YPD; 20 g/L dextrose, 10 g/L yeast extract, 20 g/L peptone) in a 96-well V-bottom plate covered by breathable rayon sealing film (Nunc #241205) in a 30˚C shaking incubator set to 300 rpm. The liquid cultures were briefly resuspended by pipetting and arrayed in a 384-position configuration on CM plates (2% w/v agar, 50 mL/plate) using a ROTOR HDA (Singer Instruments). These plates were grown for 2 days at 30˚C, after which they were arrayed to CM plates using 384 short pin pads. After a total of 3 passages on CM plates, the colonies were pinned on CM with either copper (0 to 40 µM) or sodium (0 to 1200 mM). These plates were photographed daily for 4 days following pinning in a PhenoBooth running PhenoSuite v. 2.20.504.1 (Singer Instruments) at 2560 X 1920 pixel resolution under default lighting settings.

A linearized relationship between colony size and cell number was established to calculate Malthusian growth rates from changes in population size (Fig 1A). Study strains were pinned on CM plates in 96, 384 and 1536 -position configurations in order to obtain colonies that

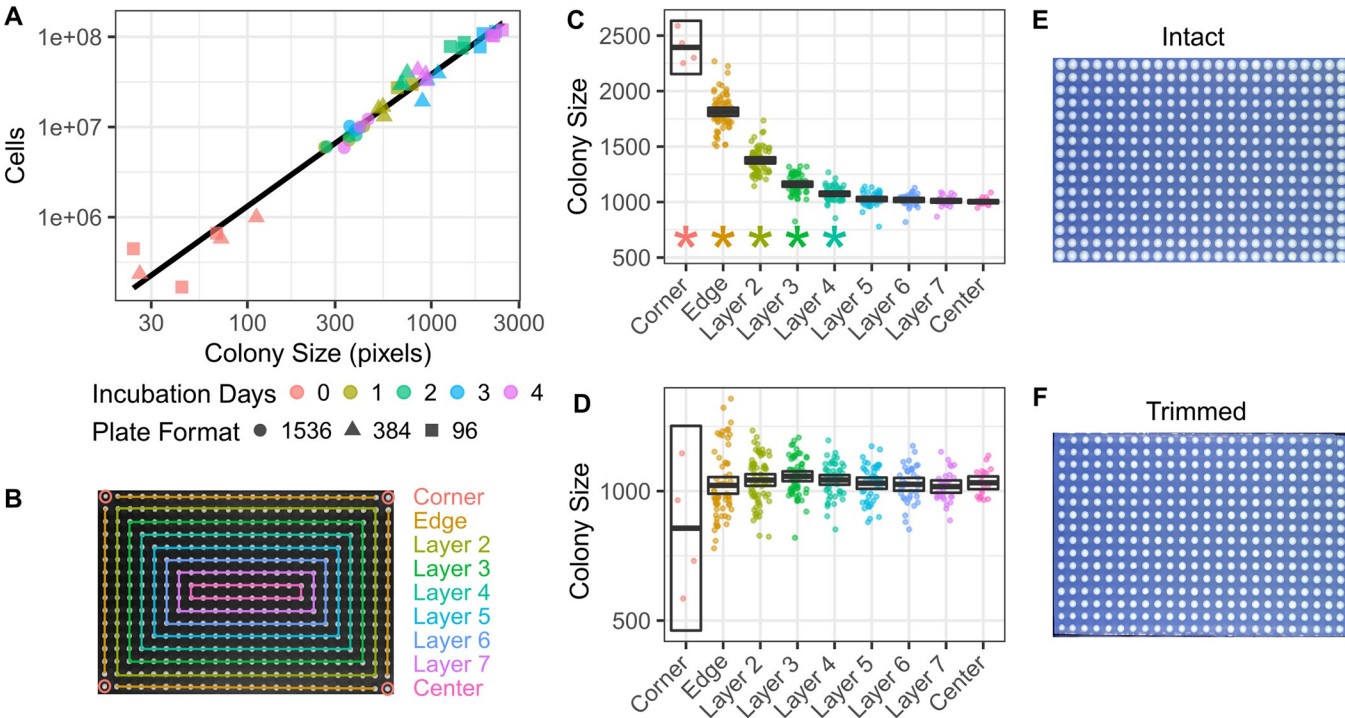

**Fig 1. Estimating the number of cells per colony and minimizing edge effects.** The relationship between cell number and colony size was best represented by the regression (black line) of log cells on log colony size for colonies grown at three different plate format densities and measured over four days of growth (A). Colony positions on the plate were modeled as a series of concentric layers with matched numbers of neighbors along the edges and corners (B). Colony sizes (in pixels) were quantified after 4 days growth with the edges either left intact (C) or trimmed (D). Boxes indicate the 95% confidence interval bisected by the mean. Stars indicate layers that are significantly different from the center (FDR < 0.05, two sample t-test). Photographs of the plates corresponding to the colony size plots for the intact and trimmed plates in panels C and D (E and F, respectively). All panels are from growth on CM plates.

reflected the range of sizes seen in the phenotyping study. Once a day for four consecutive days, the pinned plates were imaged and three random colonies were excised from each plate and resuspended in 1 mL 0.1% (w/v) peptone water [36]. The resuspended colonies were serially diluted with peptone water and enumerated in a Bright-Line Hemacytometer. A linear regression was fitted to the log-transformed size and cell measurements to derive the following relationship: $\log10(Cells) = \log10(Size)*1.46397+3.19251$. The average day 0 colony size (57 pixels) was determined from the 96 and 384-pinned plates due to both pads sharing the same pin head dimensions. This starting colony size and its corresponding cell number (579,506 cells, inferred from the regression) was set as the day 0 size for all later analysis. Colonies that were either smaller than 57 pixels or had morphological characteristics outside PhenoSuite's settings (e.g. colonies that grew into each other) were set to 0 pixels by the software's default behaviors. These data were flagged and then visually examined to be either re-incorporated into the data at the day 0 size if they were small colonies, or discarded if the colonies were absent or morphologically complicated.

Three strategies were used to address colony size variation originating from neighboring colonies and/or proximity to the perimeter, frequently referred to as "edge effects" [37–40]. First, strain replicates were arrayed to plate locations in a manner that would randomly assign strains to their neighboring colonies while still providing each strain with a level of representation at each layer with respect to plate perimeter proximity (Fig 1B). This was achieved by arraying each group of strain replicates in a circular snaking pattern starting at the corners and moving inward with a randomized strain order. Second, the agar from the plate edge to 3 mm of the outermost pinned colonies was removed using a flame-sterilized scalpel immediately following pinning. A parallel set of plates were made on CM with NaCl ranging from 0 to 1200 mM without subsequent trimming. Colony sizes on trimmed and intact plates were measured for 4 consecutive days following pinning and examined as a function of distance to the edge of the plate (Figs 1C–1F and S1). Corner colonies were removed from the data due to their pronounced variance on trimmed plates. Third, all colonies were normalized to row and column plate-wide means.

Fitness was estimated from row and column mean-normalized colony sizes on trimmed plates. Normalized colony sizes were translated to cell numbers and per-generation Malthusian growth rates were obtained from the slopes of the linear regressions of log(cells) on the number of ancestor generations calculated using $\log2(N_i/N_0)$ where $N_i$ is the number of cells on day i (S2 Fig). The reported fitness difference, equivalent to the selection coefficient, is the difference between the slopes of each evolved replicate and the average slope of the evolved strain's ancestors on the plate.

Four parallel sets of experiments were performed to examine the reproducibility of fitness estimates based on colony size. Two sets sharing the same plating layout were independently pinned. An additional two sets were independently pinned using a different randomized order for strain locations on the plate. All four experimental sets were trimmed immediately after pinning and fitness differences were determined from colony size data that was normalized by row and column mean with the corner colonies removed. Reproducibility between experimental sets was assessed by Pearson's correlation using strain average fitness differences under each stress condition. The first set was used for all presented analyses except for assessing reproducibility, which used all four sets.

### Liquid growth assays

The growth characteristics of the evolved and ancestor strains were examined to choose sodium and copper stress conditions for competitive growth assays. Cultures (500 μL) were

grown in 2 mL deep well V-bottom 96-well plates covered with breathable rayon sealing film in a 30˚C shaking incubator at 300 rpm. For 2 consecutive days, cultures were diluted serially 1/1000 into fresh media every 24 hours. On the third day, cultures were diluted 1/1000 into 200 μL CM with various concentrations of CuSO4 (0, 4, 5, 6, 6.5, 7, 7.5 and 8 μM) or NaCl (0, 34, 68, 103, 137, 171, 222, 274 mM) in a 96-well plate. Culture OD600 was recorded every 4 minutes over 24 hours in an Epoch 2 plate reader (BioTek) set to 300 rpm and 30˚C.

Growth parameters were extracted from the recorded growth curves (S3 Fig). The maximum growth rate was determined using a 15-point (1 hr) sliding window along the recorded data plotted as log(OD600) as a function of time. Lag time was calculated from the intersection of the regression lines from the first 15 points and the maximum growth rate line. Area under the curve was calculated using the trapezoid rule with the area under the smallest measured log(OD600) removed. Carrying capacity was defined as the highest OD600 achieved by the culture within the 24 hour growth period.

## Competitive fitness assays

Competitive fitness assays were used to estimate the change in fitness of each evolved strain relative to its ancestral pair. Competitions were performed between each evolved or ancestor strain and a common YFP-expressing reference strain (YJF4604) for three consecutive days under three conditions: CM, CM + 5 μM CuSO4, and CM + 103 mM NaCl. These stress conditions were chosen based on the slight growth differences observed between evolved and ancestor strains in pure culture liquid growth assays (S3 Fig), thereby ensuring sufficient growth for measurable quantitation of all strains in competitions. Strains were grown in pure cultures overnight in CM, mixed with the YFP marked strain, and diluted 1/1000 in 500μL media in one of three stress conditions. Cultures were grown in 2 mL deep well V-bottom under the same conditions described above and were diluted 1/1000 into fresh media every 24 hours for a total of 3 days of growth. Seven replicate fitness assays were conducted for each strain.

The proportion of YFP-expressing cells was determined by flow cytometry each day. At least 20,000 ungated events were collected using an Accuri C6 (BD Biosciences) set at 14 μL/min.

FACs files were processed for gating and event extraction using R packages "flowCore" [41] and "ggcyto" [42]. Gating parameters to remove cell clumps and debris were determined from a pooled set of all runs from a particular day and stress condition (S4 Fig). First, events outside a 99% ellipse gate on a log10 side scatter area (SSC-A) vs. log10 forward scatter area (FSC-A) scatterplot were discarded. The surviving events were plotted on a forward scatter height (FSC-H) vs. width scatterplot and all events outside a 95% ellipse were discarded. YFP expression from the surviving events were evaluated using the measurements collected from the FL1 channel (excitation 488 nm, emission 533/30 nm) with gating set at the mid-point between the two largest populations observed on a log10 YFP area (YFP-A) histogram.

Fitness differences were estimated by the change in YFP proportions over time. Pure culture controls consisting of the YFP-expressing reference showed the presence of YFP-negative cells (S4D Fig) and were used to estimate the fraction (f) of YFP-negative cells in competitions. In mixed cultures, the true number of YFP cells was estimated by P/(1-f), where P is the number of YFP positive cells. After rearranging terms, the ratio of the unmarked strain to the YFP marked reference was estimated by:

$$Strain/Reference = \frac{TotalCounts(1-f)}{P} - 1 \qquad \text{Eq1}$$

Selection coefficients were estimated from the slope of the regression of log(strain/reference) ~ generation (equation 2.4 in [32]), and used to calculate fitness differences between each evolved strain and its ancestral pair. The number of elapsed generations was calculated for each ancestor strain using the change in cell concentration. Cell concentrations were calculated from the positively-identified flow events and the volume usage tracked by the Accuri C6 flow cytometer during data acquisition.

## Statistical analyses

Fitness differences were evaluated for significance using a two-sided, one-sample t-test with μ set at 0. The resulting p-values across strains and conditions were adjusted for multiple testing by the Benjamini and Hochberg false discovery rate (FDR) method. The error in measured fitness differences was assessed by examining a linear model, fitness ~ evolved strain. The root mean square error (RMSE) was calculated from the residuals of an ANOVA. Power curves for each assay were generated for two-sided, one-sample t-tests with the number of replicates performed for each assay or technical/biological replicate using the R package "pwr" [43]. RMSE was used to estimate the common error variance and calculate power at a significance level of 0.95.

## Results

To evaluate our ability to measure fitness, we selected six evolved strains and their ancestors from a previous study of adaptation to constant and fluctuating environments [34]. The evolved strains were obtained from 50 days (~500 generations) of serially passaged liquid cultures under five different treatments. The treatments consisted of an evolutionary history of no stress in CM (EH0), constant sodium (NaCl) or copper (CuSO4) stress at 80% of the ancestor's lethal limit (EH80), or daily fluctuations between no stress and either sodium or copper stress (EH0_80). Previously, the fitness of the evolved populations was measured in the presence and absence of sodium and copper stress [34]. The six strains were isolated from populations that had a range of fitness changes, from less than 1% to over 10%. For reference, we first measured the fitness of these strains using competitive fitness assays in liquid cultures.

### Competitive fitness assays

Competitive fitness assays were performed for each of the ancestral and evolved strains using a common YFP-expressing reference strain in the presence of either sodium, copper, or no stress. All the evolved strains demonstrated fitness differences from their ancestors in at least 2 of the 3 conditions (FDR < 0.05, Fig 2). Except for the strains which experienced fluctuating environments, the largest fitness gains were found in the environment in which the strain was evolved. Fitness differences as small as 0.0221, 0.005 and 0.0148 were detected by competitions in no stress, copper stress, and sodium stress, respectively (S2 Table). The root mean squared error (RMSE) of fitness is comparable to previous competitive fitness assays (0.0176, [11]) and indicates there is power to detect fitness differences as small as a few percent (S5 Fig).

### Measuring fitness from colony size

Fitness differences can also be estimated from Malthusian growth rates when measured in units of generations [32]. Malthusian growth rate, and thus fitness, can be estimated from changes in colony size if colony size is converted to population size. To determine the relationship between colony size and cell number, we robotically pinned the study strains at three plating densities (96, 384, 1536) on agar plates and measured colony sizes and cell numbers from

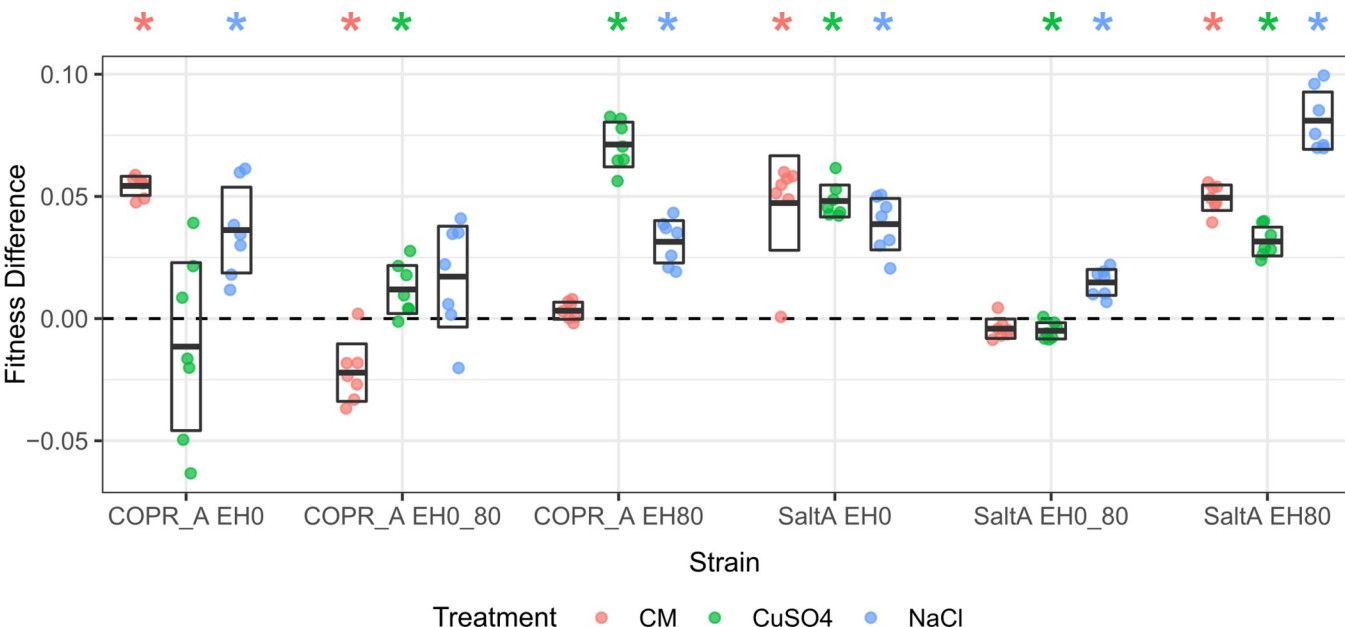

**Fig 2. Competitive fitness differences between evolved strains and their ancestors in three environments.** Strains labels are based on evolution under constant stress (EH80), fluctuating stress (EH0_80), or no stress (EH0) for both copper (COPR) and sodium (SaltA) treatments. Fitness was measured over a three day period in the presence of 5 μM copper (CuSO4), 103 mM sodium (NaCl) or without stress (CM). Boxes represent the 95% confidence interval bisected by the mean of 7 replicates. Stars indicate FDR < 0.05.

individual colonies for four consecutive days (Fig 1A). Using colonies grown for different durations and under different pinning densities provided us with a wide range of colony sizes. The relationship between colony size and cell number was found to be best linearized using the log transformation of both cell number and colony size (R2 = 0.9689), and showed that, on average, 5.79x105 cells were pinned on day 0 and expanded to 3.79x107 cells after four days growth in 384 colony format on CM plates.

Colony size can exhibit edge effects due to the number and size of neighboring colonies. On CM agar plates, where colony sizes for all the study strains were fairly similar, we found strong edge effects as measured by the correlation between a colony's size and its proximity to the edge of the plate (Fig 1B, 1C and 1E). Edge effects increased over time and extended 4 layers deep by the second incubation day (two sample t-test, FDR < 0.05, S1 Fig). Removing the agar to within 3 mm of the perimeter colonies directly after pinning greatly reduced the edge effects as measured by size differences between layers (Figs 1C–1F and S1). Corner colonies sizes were found to be highly variable compared to other layers despite trimming and were excluded from subsequent analysis.

To further characterize edge effects, we pinned a single strain in 96, 384 and 1536 colony format on CM plates and measured the amount of variation in colony size explained by layer for trimmed and intact plates over four days (S6 Fig). For intact plates, edge effects increased over time at all three colony densities, but the 96 colony format increased the most. This observation is consistent with a saturation of edge effects in the higher density formats where nutrient limiting conditions for all colonies are reached more rapidly. In comparison to intact plates, the trimmed plates showed greatly reduced edge effects for both the 384 and 1536 colony formats. For the 96 colony format, trimming increased edge effects because trimming resulted in smaller colonies at the edges compared to the center.

Colony size normalization schemas have been previously described to control for edge effects [30, 37, 38,44, 45]. We tested the effects of normalization by row and column mean or

median, as well as by the layer mean on intact and trimmed CM plates (S7 Fig). We found that row and column normalization to the mean greatly improved the variance in colony size explained by strain for the intact plate (R2 of 0.017 to 0.291), but also improved it for the trimmed plate (R2 of 0.159 to 0.451). We extended our analysis of edge effects to trimmed and intact plates with different sodium concentrations and to trimmed copper plates (S8 Fig). Normalization to the row and column mean increased the variance in colony size attributable to strain for both trimmed and intact plates. At lower sodium concentrations, colony size phenotypes attributed to strain on trimmed plates matched or exceeded intact plates likely due to the effects of neighboring colonies rather than chemical stress being the predominant cause for colony size limitations. However, trimming decreased the amount of variance explained by strain at the uppermost sodium concentrations. Under these conditions, colony sizes for all strains were small and may have been more influenced by slight errors in the trimming procedure. Row and column normalization increased the variance explained by strain for all sodium plates but decreased it at high concentrations of copper. Normalization can thus decrease variance explained by strain when there are extreme colony size differences and strains with different sizes are not evenly distributed among rows and columns.

## Fitness measurements based on colony size

The fitness of the evolved strains relative to their respective ancestors was measured over 5 copper concentrations (0 to 40 μM) and 13 sodium concentrations (0 to 1200 mM) on trimmed plates using 32 replicates per plate and with row and column mean colony size normalization. The evolved strains showed fitness differences from their ancestors under one or more conditions, with the largest fitness gains occurring in the stress under which the strain was evolved (FDR < 0.05, Fig 3). One exception was that the strains evolved in the absence of stress (EH0) displayed reduced fitness in the absence of either stress. Fitness differences as low as 0.0024 and 0.0026 were detected on sodium and copper plates, respectively (S2 Table).

We tested the reproducibility of measuring fitness from colony sizes using independent experiments with either identical or different pinning layouts (S9 Fig). Mean fitness changes were highly reproducible with better replication between experiments sharing the same plating layout than those with different layouts (Pearson's r = 0.992–0.993 versus 0.974–0.986, respectively), indicating some residual position effects that were not eliminated by trimming and row column normalization.

## Comparing colony size and competitive fitness assays

The power to detect fitness differences depends on the sample size and the experimental error. Using RMSE to measure the average error in fitness measurements, we found the colony size assays had lower error compared to the competitions at low sodium and copper concentrations, but higher error as stress concentrations increased (S2 Table). The larger sample sizes for the colony assays compared to the competitive assays (32 versus 7) yielded higher power for all but the highest stress concentrations (S5A, S5C and S5E Fig). When the power of the two assays was compared for an equivalent number of replicates, colony size was only more powerful at lower stress concentrations (S5B, S5D and S5F Fig).

Even though both fitness assays measured the effects of sodium and copper stress, fitness on solid agar and in liquid cultures may differ for a number of reasons. We examined the correlation between competitive fitness and fitness based on colony size (Fig 4). The correlations in fitness were highest for mid-ranged sodium and high copper concentrations. However, many were negative for high sodium and low copper concentrations and in the absence of both stresses. Thus, without strong stress the two assays represent substantially different

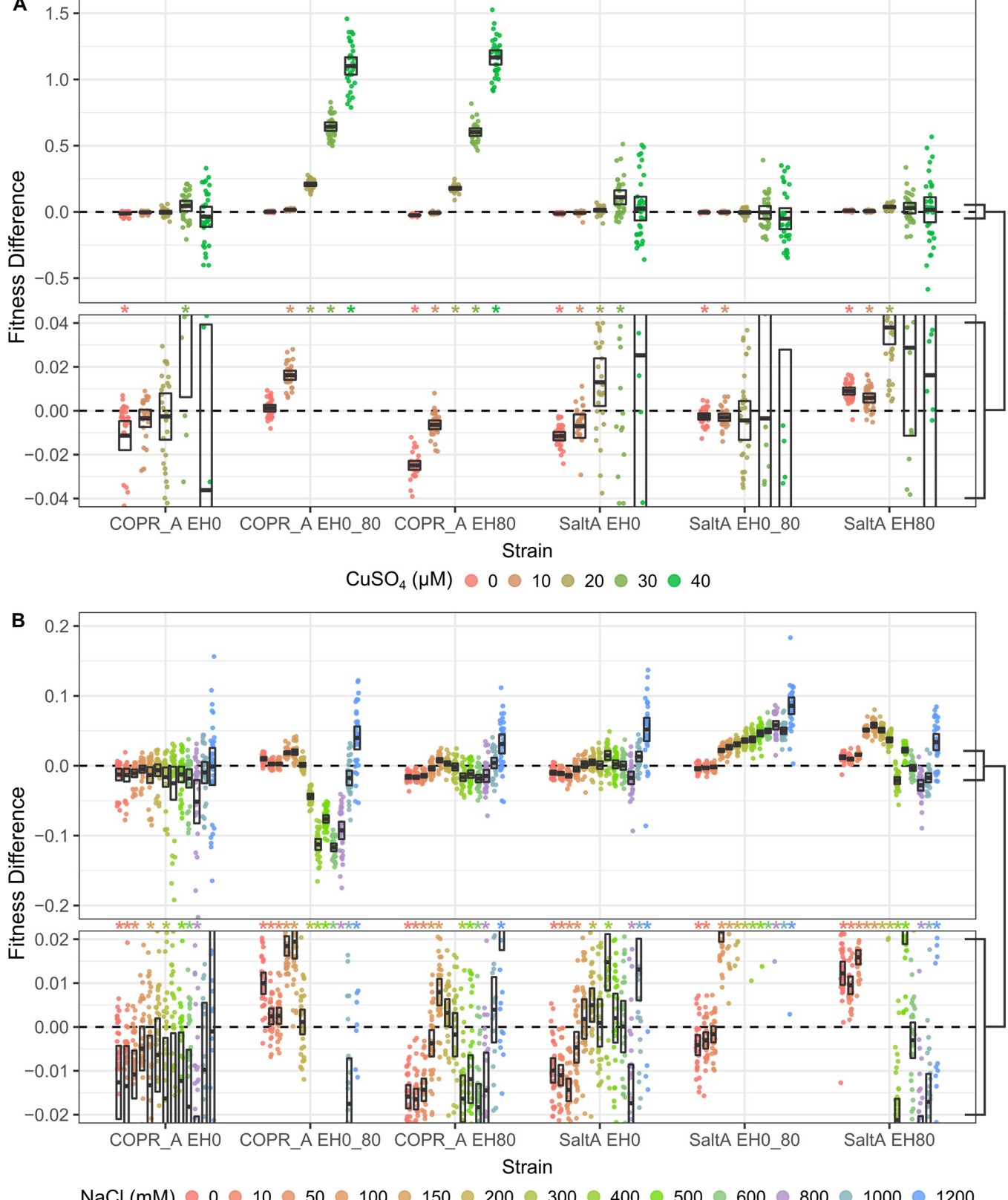

**Fig 3. Fitness differences between evolved strains their ancestors determined from colony sizes.** Fitness was measured on CM plates with varying concentrations of copper, (A, zoomed below) or sodium (B, zoomed below) over a four day period. Boxes represent the 95% confidence interval bisected by the mean of the replicates. Stars indicate FDR < 0.05.

aspects of fitness. Even under the conditions with the highest fitness correlation the rank order of strain fitness differed between the two assays (Fig 4B and 4C).

## Comparing colony size and fitness from colony size

Because colony size is a commonly used measure of fitness, it is practically useful to know how it is related to per generation measurements of relative fitness based on colony size. A linear

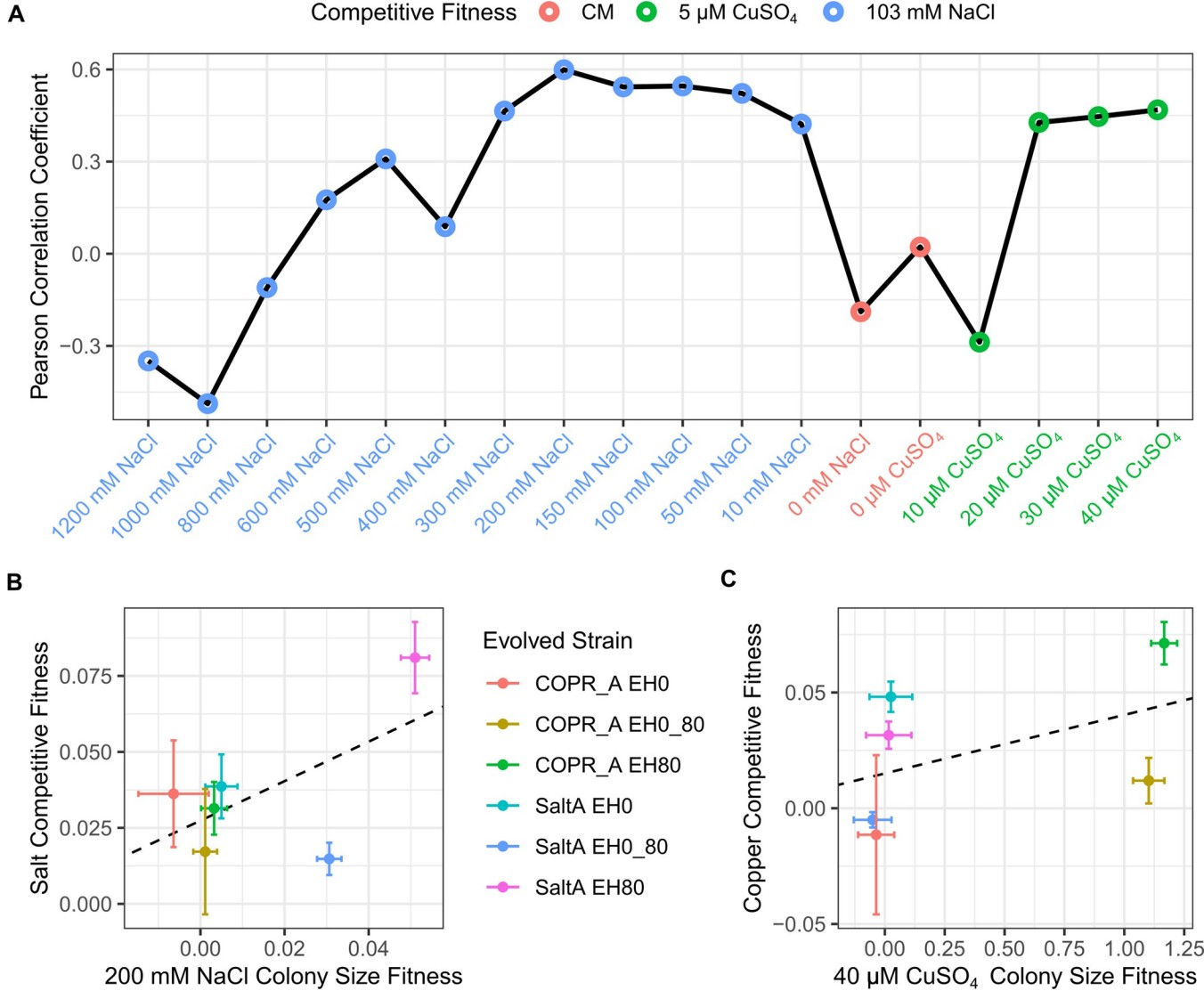

**Fig 4. Comparison of fitness differences derived from competitive growth and colony size assays.** Correlations between competitive fitness assays and the colony size assays with analogous stress conditions (A). Fitness differences of each evolved strain from competitive fitness and colony size assays under conditions with the highest correlation for sodium and copper stresses (B and C, respectively). Means are represented by points with 95% confidence intervals; the linear regression of the displayed means are shown by dashed lines.

relationship between the log ratio of colony size and relative fitness is expected because our measurement of fitness is based on the log difference in population size and population size is exponentiated from colony size. We find a linear relationship between fitness and the log ratio of colony sizes, although the slope changes depending on the stress (S10 Fig). The change in slope depending on concentration is most likely a consequence of fitness being measured on a per generation basis, whereby a given colony size ratio will translate into a larger relative fitness difference at higher stresses where there have been fewer generations of growth in the susceptible ancestor.

## Discussion

Colony size has been extensively used as a high-throughput and sensitive means of quantifying microbial growth [19]. In this study we estimated fitness from colony size to determine whether colony size can be used to measure small (~1%) differences in relative fitness that occur during experimental evolution. Because standard measures of relative fitness used to calculate selection coefficients are measured on a per generation basis [32], the translation between colony size and fitness is derived from estimates of population size and the number of elapsed generations. Using this translation, we find that the experimental error in estimates of fitness based on colony size is comparable to that obtained from widely used competitive growth assays in liquid medium. Below, we discuss both the strengths and weaknesses of measuring fitness from colony size and conclude that it offers a compelling advantage across a variety of experimental circumstances.

The availability of robotic pinning machines [46] led to the widespread use of colony size to measure microbial growth from images of gridded colonies on agar plates as well as the development of methods to extract precise and reproducible measures of colony growth [23–26, 28–31]. In addition to simple measures of colony size based on area, colony growth can be measured from continuous imaging over time [29, 31], and some methods incorporate colony opacity or phloxine B staining [28, 30]. However, previous estimates of fitness used relative colony size or relative growth rate, which should be correlated to but not equivalent to per generation measures of fitness [32]. Our goal was to assess whether the error in per generation fitness differences based on colony size is small enough to detect the 1% differences typically found in experimental evolution. We found the RMSE of per generation fitness differences based on colony size to be similar to that based on competitive growth assays (under 2% for most conditions, S2 Table). If we assume error scales linearly with fitness and fitness is scaled to one, our RMSE in fitness can be compared to prior estimates of experimental error based on the coefficient of variation (CV). Using Scan-o-matic, CV was estimated to be 2% for the maximum doubling time using 20 minute imaging intervals [28]. Using *pyphe*, a similarly low CV was estimated for *S. pombe* colony size endpoints and maximum growth rates [30]. Finally, LI Detector found a CV of ~4% for fitness estimated from 11 imaging timepoints [31] and the average CV of colony size from single gene deletions was 1.6% [37]. Thus, despite differences in experimental design, most measures of fitness based on colony size appear well suited to detect small per generation fitness differences found in experimental evolution studies.

Positional effects are a major disadvantage to phenotyping colonies pinned at high density. These are most apparent at the edges of a plate where colonies grow larger than replicate clones located at the interior. Additionally, colony sizes are also influenced by the sizes of neighboring colonies, e.g. [30, 37]. Consistent with this explanation, we observed the strongest edge effects at the highest colony density (1536) and on the plates that had grown for the longest time period (S1 Fig). This makes compensating for positional effects by normalization difficult or inaccurate if only a small number of colonies form on a plate, e.g. high sodium or copper. We

found that trimming excess agar from a plate dramatically reduces but does not eliminate edge effects. To mitigate any remaining positional effects, we used a randomized snaking method for ordering strains on the plate and row column normalization for colony size corrections. Even so, not all positional effects were eliminated because we observed a higher correlation between replicates pinned with the same positional layout compared to those pinned with a different layout. Other approaches to dealing with positional effects have used more sophisticated spatial normalization procedures, including the use of gridding or smoothed position effects, adjacent colonies, and the use of control colonies pinned to the borders of a plate or interspersed with those of interest [25, 28, 30, 31, 46, 47]. Although trimming reduced spatial effects (S7 Fig), the improvement was modest with row and column normalization and it seems reasonable to suppose that more sophisticated normalization procedures could yield results similar to or better than agar trimming. However, any spatial normalization procedure is likely to be challenged by conditions where most strains, including controls, do not grow.

Accurate estimates of fitness depend on the time interval over which changes in population size are measured. We found the lowest error when fitness was measured from daily imaging over a four day period. The average RMSE over all conditions for the experimental replicate shown in S2 Table and Fig 3 was 0.0469, 0.0410, 0.0390 and 0.0350 for fitness differences derived from colony size data spanning day 0 to days 1, 2, 3 and 4, respectively. This result is consistent with previous studies which found that variance in colony size decreased and then increased over time and that population growth rates based on colony size had lower error than endpoint measurements [28, 30]. One notable source of error is the variance in the number of pinned cells (CV = 0.584 for 96 pins and 0.632 for 384 pins) [28]. Despite this variation, colony size among replicates is quite uniform after several days of growth. One potential explanation is that the initial colony size depends more on the area cells are pinned to rather than their density due to colony growth dynamics. Colony size depends on cell divisions at the edge of the colony [48, 49], with internal cells ceasing cell division and entering stationary phase [50]. If pinned cells are close to the initial carrying capacity of the pinned area [24] then differences in the initial density may play a relatively small role in determining final colony size.

Fitness based on competitive growth assays and colony size are both estimated by changes in population size relative to a reference but they differ from one another in a number of important ways. Both assays measure fitness as an average over time and cells. Similar to liquid competition assays [51], we observed different estimates of fitness depending on the time interval of growth (S2 Fig). Both assays also measure the average fitness of cells in the population, but cells within colonies are more heterogeneous in their rates of division. Consequently, differences in relative fitness of actively dividing cells in a colony are likely larger than the average differences that we measured. However, it is important to recognize that there is also substantial cell heterogeneity in liquid cultures [52]. In addition to measuring average fitness, the two assays are also similar in that they typically use the generation time of the reference strain to estimate per generation differences in relative fitness. As previously pointed out [32], per generation measurements of fitness are comparable across studies and directly relevant to evolutionary dynamics. However, both fitness assays are complicated by density- and frequency-dependent selection and selection on variable generation times [53].

What is the best measure of fitness? A primary advantage of using colony size is that it facilitates the use of unmarked strains and the capacity to measure fitness across many conditions simultaneously. In a single condition, competitive fitness assays of barcoded strains in liquid culture have the highest throughput [54]. However, the two assays are not equivalent because growth in liquid and solid medium may differ due to oxygen levels or other aspects of the environment. We find that while they qualitatively agree on which strains are the most copper and sodium resistant, there is little or no correspondence between the two in the absence of stress

(Fig 4). One interpretation of this outcome is that adaptation to stress tends to provide benefits in other environments with the stress, whereas adaptation in the absence of stress are more specific to the culture environment. A prior study of adaptation to liquid batch cultures showed that most of the gains in fitness could be attributed to benefits that accrued during respiration but were only realized during the lag phase of the following growth cycle [51]. Trade-offs in fitness have also been found between fermentation, respiration and stationary phase [51, 55]. Thus, one explanation for the difference between liquid and solid medium fitness assays is that the solid assays did not include multiple cycles of growth or had a different composition of growth phases. However, our sample size was small (six evolved strains) and prior studies with larger numbers of strains that ranged from small to large fitness effects found correlations between solid and liquid based measures of fitness of 0.78 [28] and 0.53–0.70 [37]. We conclude that fitness measurements derived from colony size differ from those derived from competitive fitness assays, and both can provide useful measures of this critical evolutionary parameter in microbial populations.

## Supporting information

**S1 Table. Strains used in this study.** Evolved strains are indicated by the name of their stress (COPR_A for copper and SaltA for sodium), and the percent lethal limit in which they were raised (*e.g.* EH0_80 indicates an evolutionary history of environment fluctuation between 0 and 80 percent lethal stress).
(PDF)

**S2 Table. Summary of measurement properties from the competitive fitness and colony size assays.** The root mean square error (RMSE) and variance in fitness explained by strain ($R^2$) are derived from the fitness difference ~ evolved strain relationship at each stress concentration using the data shown in Figs 2 and 3. Minimum detectable fitness differences is from power analysis at 0.80 power and 0.95 significance.
(PDF)

**S3 Table. Reagents and equipment used in the experiments.**
(PDF)

**S1 Fig. Edge effects over time.** Colony photographs and size distributions by layer for days 1–3 associated with Fig 1B–1E. Evolved and ancestor strains were arrayed in a randomized pattern on CM agar plates either left intact or trimmed immediately after printing (A and B, respectively). Boxes indicate the 95% confidence interval bisected by the mean. Stars indicate layers that are significantly different from the center (FDR < 0.05, t-test).
(PDF)

**S2 Fig. Estimating fitness differences from colony size.** Examples of colony size, estimated number of cells, generations and fitness differences over four days with corresponding photographs (A). Colony size areas (pixels) were recorded for four consecutive days after pinning and the size on day 0 was set to 57 (B). Colony size was converted to cell number using the experimentally derived log-log relationship (Fig 1A) and plotted as a function of ancestor generations (C). Fitness differences were determined from the difference in regression slopes (from panel C) between each evolved strain and the average of its ancestor pair as a function of elapsed days (D).
(PDF)

**S3 Fig. Liquid growth curve characteristics of evolved and ancestor strains.** Pure cultures were used to determine the single stress concentrations for competitive growth assays. The

area under the curve, carrying capacity, time lag and maximum growth rate for different concentrations of copper and sodium were determined from growth curves of each strain (n = 1). Growth measures are shown relative to growth in CM without added copper or sodium. Dotted lines indicate the concentration chosen for competitive fitness assays.
(PDF)

**S4 Fig. Gating scheme used to measure the proportion of YFP cells for the competitive fitness assays.** Gating parameters were determined from the pooled events of all runs pertaining to a single day and stress. The first gate excluded events outside the 99% ellipse from log10 SSC-A (side scatter area) vs. log10 FSC-A (forward scatter area)(A). This was followed by event exclusion from a 95% ellipse on a FSC-H (forward scatter height) vs. width (B). The YFP gate was manually set at the midpoint between the two largest populations on a log10 YFP-A histogram and kept at a constant value throughout all experiments (C). Example YFP-A histograms after gating showing the competition between an ancestor strain and the YFP-expressing reference strain, and a reference-only control consisting of YJF4606 alone (D). Note the small proportion of the events exhibited fluorescent signalling below the YFP gate in these reference only controls (1–19%) which are accounted for as the false negative fraction (Eq 1).
(PDF)

**S5 Fig. Power analyses of competitive fitness and colony size assays.** Power and number of replicates needed for competitive fitness (dashed lines, labeled "C.F.") and colony size assays (solid lines) for CM only (A and B) and various concentrations of copper (C and D) and sodium (E and F). Power as a function of fitness difference was derived from each assay's measured performance (A, C and E). The number of replicates needed to achieve 0.8 power at 0.95 significance is plotted as a function of fitness difference (B, D and F).
(PDF)

**S6 Fig. The proportion of variance explained by layer as a function of incubation time.** Variance explained ($R^2$) is shown for intact and trimmed plates arrayed on CM plates in 96, 384 or 1536 density formats. All colonies are of ancestor strain YJF4679 and corner colonies were excluded.
(PDF)

**S7 Fig. Colony sizes from various normalization methods.** Colonies are after 4 days growth from trimmed and intact CM plates with no normalization and normalization by row and column mean, row and column median, or layer. Numbers in the upper right indicate the size variance explained by strain (colony size ~ strain). Boxes show the 95% confidence interval bisected by the mean with stars indicating layers that significantly differed from the center layer (FDR < 0.05, two sample t-test).
(PDF)

**S8 Fig. Colony size variance explained by strain after normalization.** Variance explained by strain ($R^2$) before and after row and column mean normalization for either intact or trimmed plates containing various concentrations of sodium (left) or trimmed plates with copper (right).
(PDF)

**S9 Fig. Fitness differences estimated from independent experiments with either same or different pinning layouts.** Points in black indicate that independent experiments resulted in overlapping 95% confidence intervals whereas points in red did not. Green boxes indicate the locations of the insets shown in each plot, the y = x line is shown in black.
(PDF)

**S10 Fig. Comparison of fitness differences and colony size ratios.** The log of colony size ratios from Fig 3 were calculated by dividing the colony size of each evolved replicate by the average size of its respective ancestors on copper (A) and sodium (B) plates. The location of the inset is indicated by the black box.
(PDF)

## Acknowledgments

This work was made possible by access to the Center for Advanced Research Technologies' Flow Cytometry Core at the University of Rochester. We also thank members of the Fay lab for their comments and suggestions.

## Author Contributions

**Conceptualization:** Emery R. Longan, Justin C. Fay.

**Formal analysis:** James H. Miller.

**Funding acquisition:** Justin C. Fay.

**Investigation:** James H. Miller.

**Methodology:** James H. Miller.

**Resources:** Vincent J. Fasanello, Ping Liu, Carlos A. Botero.

**Visualization:** James H. Miller.

**Writing – original draft:** James H. Miller, Justin C. Fay.

**Writing – review & editing:** Vincent J. Fasanello, Emery R. Longan, Carlos A. Botero.

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
