## [Decision Letter · Decision Letter 0]

25 Jul 2022

PONE-D-22-19057Using colony size to measure fitness in Saccharomyces cerevisiaePLOS ONE

Dear Dr. Fay,

Thank you for submitting your manuscript to PLOS ONE. After careful consideration, we feel that it has merit but does not fully meet PLOS ONE’s publication criteria as it currently stands. Therefore, we invite you to submit a revised version of the manuscript that addresses the points raised during the review process.

Specifically, the reviewer #1 has several technical questions. The reviewer #2 thinks that the you can improve the manuscript by comparing with the previous similar methods. Like these two reviewers, I also think that the manuscript presented in an intelligible fashion so that I look forward to seeing the revised manuscript shortly.  ==============================

We look forward to receiving your revised manuscript.

Kind regards,

Yoshikazu Ohya, PhD

Academic Editor

PLOS ONE

Journal Requirements:

3. Please include a copy of Table 2 which you refer to in your text on page 12.

Reviewers' comments:

Reviewer's Responses to Questions

**Comments to the Author**

1. Is the manuscript technically sound, and do the data support the conclusions?

Reviewer #1: Partly

Reviewer #2: Yes

2. Has the statistical analysis been performed appropriately and rigorously? 

Reviewer #1: Yes

Reviewer #2: Yes

3. Have the authors made all data underlying the findings in their manuscript fully available?

Reviewer #1: Yes

Reviewer #2: Yes

4. Is the manuscript presented in an intelligible fashion and written in standard English?

Reviewer #1: Yes

Reviewer #2: Yes

5. Review Comments to the Author

Reviewer #1: In the present manuscript, Miller et al. have reported colony size as a metric to measure differences in relative fitness. There must be a broad audience for such a study given its high-throughput nature. However, there are some points which need to be addressed:

Major points

• Authors checked fitness through colony size using an image processing method. Do the authors only consider pixel count? since (presumably) images were taken in 2D space, pixel intensity could have improved the robustness of the methodology.

• Authors have run very different number of replicates for colony size and competitive fitness (32 vs 7). This is a huge difference. I also do not understand what n <= 32 (line 319) and ~32 (S2 Table) mean. It seems there were different number of replicates in colony size experiments. These differences can affect the power (which is one of the author’s metric to evaluate and compare their results)

• Was building the linear regression model (log10(Cells)=log10(Size)*1.46397+3.19251) before the mean of row and column normalization or after? If it is before, and the 3 randomly selection occurred everyday (day 1 to day 4), it means that edge effect played a big role into your model.

• How would authors explain variability in RMSE and R2 values (S2 Table)? What is the biological reason. As you would see in following plot, there is no trend:

See the attached figure

• As authors mentioned, number of replicates influence power of a statistical model (line 315). The observed power in this manuscript could also be due to their experimental design (large number of replicates; 32).

I also could not think of any reasons to justify power by levels of stress conditions (lines 322-322).

• Fig. 4. Authors linked fitness assays of solid and liquid cultures using Pearson’s r while the variability of the results (from no correlation to relative correlation and negative correlation) makes it hard to believe there is any relations.

In B and C sections, compared stress levels are different (NaCl: 200 vs 103 and CuSO4: 40 vs 5) and accordingly it is hard to make any conclusions.

• In abstract, it is stated that “colony size is as sensitive as competitive fitness assays grown in liquid medium”. There is no explanation in the main text and no evidence to support this claim.

Minor points

• To increase reproducibility, I strongly recommend the authors to provide a supplementary table mentioning all available information about chemicals and equipment used in their study.

• I was wondering if authors tried Poisson regression (instead of log10 transformation) given the data-type (cell count and pixel count).

• Supplementary figures (such as Fig.1 and S1): the FDR threshold can be inferred to be 5% but it’s better to be clearly mentioned (FDR = 5% instead of FDR-corrected). Is it one- or two-sided t-test?

• S1 Fig. A: The “intact” plates are inappropriately cropped. First column from right is cut out.

• “Various concentrations of CuSO4 or NaCl” is mentioned several times in the text but there is no reference to S2 Table

• Line 255: The sentence “Previously, the fitness of the evolved populations was measured in the presence and absence of sodium and copper stress.” needs a reference.

• The word “Stress” in legend of Fig. 2 causes confusion: “CM means no stress”.

• S8 Fig. Why there is no intact condition for cooper.

Reviewer #2: The authors report that by systematically obtaining the less significant fitness differences of "evolved strains" in solid medium and competitive culture, the measurement of growth (fitness) by colony size is sensitive enough to be comparable to competitive culture, which is generally considered a sensitive fitness test.

The authors have performed several interesting experiments. These include showing that removing the periphery of solid medium can reduce the peripheral effect, counting the number of cells in spotted colonies at various densities to determine the relationship between colony size and cell number, and the data obtained from these experiments are valuable.

On the other hand, the current format of the paper is limited to an introduction of the experimental methods performed by the authors. This is not enough to add novelty or scientific value to the paper. Most notably (as the authors mention in their paper), there have been numerous comparisons of fitness measures in colony size and other fitness tests ( especially Baryshnikova 2010), and the differences between those methods and studies and the authors' method are not discussed. The difference between those methods and studies and the authors' method has not been discussed.

In particular, as mentioned in Baryshnikova 2010, various corrections are made in the SGAtool. How does this differ from the authors' method? Is there any advantage to the authors' method? A full comparison and discussion of this would be necessary.

Minor comment:.

1) The results of individual liquid cultures, described in the methods, are not used in the main text. Why is there no mention of the difference in fitness between individual liquid cultures and colony size measurements?

2) The concentration of copper used in the "copper tolerance" test is only 1/1000th of that used in the previous report (PMID: 15059259). Where does this difference come from?

6. PLOS authors have the option to publish the peer review history of their article (what does this mean?). If published, this will include your full peer review and any attached files.

Reviewer #1: No

Reviewer #2: No

---

## [Author Response · Author response to Decision Letter 0]

23 Aug 2022

Specifically, the reviewer #1 has several technical questions. The reviewer #2 thinks that the you can improve the manuscript by comparing with the previous similar methods. Like these two reviewers, I also think that the manuscript presented in an intelligible fashion so that I look forward to seeing the revised manuscript shortly. 

Response: We have answered Rev #1 questions and have updated the manuscript as needed. We have now clearly distinguished our method from prior methods as requested by Rev #2. A detailed response and list of changes is in the response to reviewers.

---

## [Decision Letter · Decision Letter 1]

15 Sep 2022

Using colony size to measure fitness in Saccharomyces cerevisiae

PONE-D-22-19057R1

Dear Dr. Fay,

We’re pleased to inform you that your manuscript has been judged scientifically suitable for publication and will be formally accepted for publication once it meets all outstanding technical requirements.

Kind regards,

Yoshikazu Ohya, PhD

Academic Editor

PLOS ONE

Additional Editor Comments (optional):

Reviewers' comments:

Reviewer's Responses to Questions

**Comments to the Author**

1. If the authors have adequately addressed your comments raised in a previous round of review and you feel that this manuscript is now acceptable for publication, you may indicate that here to bypass the “Comments to the Author” section, enter your conflict of interest statement in the “Confidential to Editor” section, and submit your "Accept" recommendation.

Reviewer #1: (No Response)

Reviewer #2: All comments have been addressed

2. Is the manuscript technically sound, and do the data support the conclusions?

Reviewer #1: Yes

Reviewer #2: Yes

3. Has the statistical analysis been performed appropriately and rigorously? 

Reviewer #1: Yes

Reviewer #2: Yes

4. Have the authors made all data underlying the findings in their manuscript fully available?

Reviewer #1: Yes

Reviewer #2: Yes

5. Is the manuscript presented in an intelligible fashion and written in standard English?

Reviewer #1: Yes

Reviewer #2: Yes

6. Review Comments to the Author

Reviewer #1: Authors have mainly addressed my questions however one of my questions was chopped off in the response letter and their answer is irrelevant to what I had asked. I needed to know “when” they collected the “size” data (i.e., the X variable) for the model. I was wondering if it was before or after correcting for the edge effect (the three methods).

I disagree with the authors on R^2 values. To me, there is no trend in R^2. I’d guess the variability in R^2 indicate there is not enough data to analyze in higher doses; as the authors also mentioned, most of the strain could not even grow in higher doses.

Reviewer #2: The author has responded appropriately to my comments and has incorporated them into the paper.

I believe the paper has been improved and is appropriate for publication in PLOS one.

7. PLOS authors have the option to publish the peer review history of their article (what does this mean?). If published, this will include your full peer review and any attached files.

Reviewer #1: No

Reviewer #2: No

---

## [Editor Report · Acceptance letter]

3 Oct 2022

PONE-D-22-19057R1 

Using colony size to measure fitness in *Saccharomyces cerevisiae*

Dear Dr. Fay:

I'm pleased to inform you that your manuscript has been deemed suitable for publication in PLOS ONE. Congratulations! Your manuscript is now with our production department. 

Kind regards, 

on behalf of

Dr. Yoshikazu Ohya 

Academic Editor

PLOS ONE